# Redox gated polymer memristive processing memory unit

Bin Zhang[1], Fei Fan[1], Wuhong Xue[2,3,4], Gang Liu[2,3], Yubin Fu [5], Xiaodong Zhuang [2,5], Xiao-Hong Xu[4], Junwei Gu[6], Run-Wei Li[3] & Yu Chen[1]

Memristors with enormous storage capacity and superior processing efficiency are of critical importance to overcome the Moore's Law limitation and von Neumann bottleneck problems in the big data and artificial intelligence era. In particular, the integration of multi-functionalities into a single memristor promises an essential strategy of obtaining a high-performance electronic device that satisfies the nowadays increasing demands of data storage and processing. In this contribution, we report a proof-of-concept polymer memristive processing-memory unit that demonstrates programmable information storage and processing capabilities. By introducing redox active moieties of triphenylamine and ferrocene onto the pendants of fluorene skeletons, the conjugated polymer exhibits triple oxidation behavior and interesting memristive switching characteristics. Associated with the unique electrochemical and electrical behavior, the polymer device is capable of executing multilevel memory, decimal arithmetic operations of addition, subtraction, multiplication and division, as well as simple Boolean logic operations.

[1] Key Laboratory for Advanced Materials, Institute of Applied Chemistry, School of Chemistry and Molecular Engineering, East China University of Science and Technology, Shanghai 200237, China. [2] School of Chemistry and Chemical Engineering, Shanghai Jiao Tong University, Shanghai 200240, China. [3] CAS Key Laboratory of Magnetic Materials and Devices, Ningbo Institute of Materials Technology and Engineering, Chinese Academy of Sciences, Ningbo, Zhejiang 315201, China. [4] Key Laboratory of Magnetic Molecules and Magnetic Information Materials of Ministry of Education, School of Chemistry and Materials Science, Shanxi Normal University, Linfen, Shanxi 041004, China. [5] Center for Advancing Electronics Dresden (cfaed) & Department of Chemistry and Food Chemistry, Technische Universität Dresden, Dresden 01062, Germany. [6] Shaanxi Key Laboratory of Macromolecular Science and Technology, Department of Applied Chemistry, School of Science, Northwestern Polytechnical University, Xi'an, Shaanxi 710072, China. These authors contributed equally: Bin Zhang, Fei Fan, Wuhong Xue. Correspondence and requests for materials should be addressed to G.L. (email: liug@nimte.ac.cn) or to Y.F. (email: yubin.fu@tu-dresden.de) or to Y.C. (email: chentangyu@yahoo.com)

nformation storage and processing comprise fundamental functionalities of modern computer systems. With the exponential increase in digital communication of the big data and artificial intelligence era[1], computing with enormous storage capacity and superior processing speed are of great demand for applications such as market trend analysis, real-time image processing, machine-learning and etc.[2–6]. Unfortunately, this encounters two technical obstacles at the moment. On one hand the downscaling campaign of the CMOS platform is decelerating, as the quantum uncertainty at 2–3 nm makes electron behavior unpredictable, raising difficulties in continuously increasing the chips' capacity via device miniaturization[7]. On the other hand, shuttling data between the memory hierarchy and central processing unit incurs substantial latency to the von Neumann paradigm, sometimes even stalling of the computation system when handling complex tasks, e.g., optimization problems[8,9]. Physical implementation of advanced electronic gadgets with multilevel storage capacity is thus an urgent aspiration to bypass the device dimension limitation and extend the Moore's law further. Integration of computing and memory functions into a single component is also an arduous challenge faced by global microelectronic community to eliminate the frequent yet encumbering data transfer through the von Neumann bottleneck.

The latest addition of memristor into the existing electronic device list of resistor, capacitor and inductor offers opportunity for realizing high-performance neuromorphic computing[10,11]. Rather than encoding the digital data as the amount of charge stored in a transistor, memristor stores information based on the conductance/resistance change of the device in response to an applied voltage or electric field. As such, the leaking problem of the conventional transistor based integrated circuits can be efficiently eliminated when the feature size of the devices decreases to <10 nm, which greatly improves the reliability of the entire electronic system. The two-terminal structure of the memristor also favors facile integration of the devices into crossbar array and three-dimensional stack for high density applications. It is demonstrated that the redistribution and redox reaction of ionic species in oxide and chalcogenide materials can lead to continuous evolution of the device conductivity, which in turn allows the integration of information processing and multilevel storage in a single cell by emulating the physiological behavior of biological synapses[12–17]. Charge-transfer or redox phenomena can also modify the conductivity of organic analogs hugely for binary data storage applications[18–23]. In particular, the use of polymer materials benefits a lot from atomic engineering that can fine-tune electronic properties through molecular design-cum-synthesis strategy, as compared to the chaotic device performance of their inorganic counterparts[24–26]. Light weight, low-cost fabrication with solution processing and mechanical flexibility contribute additional merits to polymer electronics. It is noteworthy that although the scientific importance of the polymer memristor is clearly high, their technical potentials are still far from being fully explored, except for the simple binary memories. Theoretically, tuning the redox activity of polymer materials in an accumulation manner should be capable of endowing multilevel storage and processing operations[27–31].

Herein, we report an experimental proof-of-concept polymer memristive processing-memory unit with programmable multilevel memory and abacus functions. By introducing redox active moieties of triphenylamine (TPA) and ferrocene (Fc) onto the pendants of fluorene skeletons through Suzuki coupling polymerization and "Click" chemistry, respectively, the resultant conjugated polymer PFTPA-Fc exhibits triple oxidation behavior in both liquid electrolyte environment and solid-state thin films.

Associated with the electric field induced electrochemical reactions of the ferrocene pendant moieties, the ITO/PFTPA-Fc/Pt sandwich structure device exhibits consecutive resistive switching characteristics at either the low or high device current levels, thus executing the multilevel memory and the four basic decimal arithmetic operations of addition, subtraction, multiplication, and division, respectively. Redox gating of the device also endows it with simple binary Boolean logic operations. Interestingly, the memory and abacus functions of the polymer device are independent and inter-convertible, wherein the redox reaction of the TPA pendants leads to an abrupt bistable resistive switching at the high-voltage level and reconfigure the device between memory and processor modes. Therefore, the integration of multilevel memory and computing capability into a single memristor device through molecular design and electronic properties tuning of polymer renders an essential strategy of obtaining high-performance electronic circuits that satisfies the increasing demands of data storage and processing nowadays.

## Results

**Designing principle and synthesis of redox active polymer.** The state-of-the-art memory technology is based on binary-encoding philosophy and stores 1-bit information per unit cell. The highest chip capacity is directly limited by the resolution of the top-down lithography that defines the size and density of the device[32]. However, further refinement in the photolithography technique meets both device physical restraints and economic stresses at very small dimension[7]. Developing multilevel memories that can store more than 1-bit data per cell is thus an apparently efficient approach to increase the chip storage capacity without making compromise to the lithography extremes. The key issue in realizing multilevel storage in memristor is to achieve consecutive resistive switching characteristics, which is also crucial for the data processing operations. With this concern, our strategy herein is to incorporate multiple redox active unities into polymer materials through molecular design and synthesis, wherein their continuous redox behavior modifies the electronic structure and conductivity of the material accumulatively. To be specific, triphenylamine (TPA) and ferrocene (Fc) groups are linked as pendant onto the fluorene backbone through Suzuki coupling polymerization and "Click" chemistry, respectively (Fig. 1, Supplementary Note 1, Supplementary Figure 1, Supplementary Figure 2, and Supplementary Figure 3). TPA and Fc are well-known for their excellent redox properties and have been explored for binary memories[27,31,33–36]. The use of electron-rich building blocks of triphenylamine, ferrocene, and fluorene will nevertheless give rise to p-type semiconducting material.

The successful synthesis of PFTPA-Fc with average molecular weight of 22 kDa and a polydispersity index of 1.95 was first verified by spectroscopic, electrochemical, and electrical analysis (Supplementary Note 2, Supplementary Note 3, Supplementary Figure 4, Supplementary Figure 5, and Fig. 2). The polymer shows good solubility in toluene, tetrahydrofuran (THF), dimethyl formamide (DMF) and etc. with a concentration of 10 mg/mL, yet cannot be dissolved by either alcohol or water (Supplementary Figure 6). According to FTIR and NMR spectra of PFTPA-Fc, a 100% "Click" reaction has occurred to achieve complete grafting of ferrocene moieties onto the pendant of PFTPA-Fc, giving each repeating unit of the PFTPA-Fc polymers containing two Fc unities as shown in Fig. 1. The UV-Visible absorption spectrum of PFTPA-Fc in dilute toluene solution exhibits two absorption peaks at the wavelengths of 306 and 377 nm, respectively (Fig. 2a). The absorption peak at the shorter wavelength is attributed to the

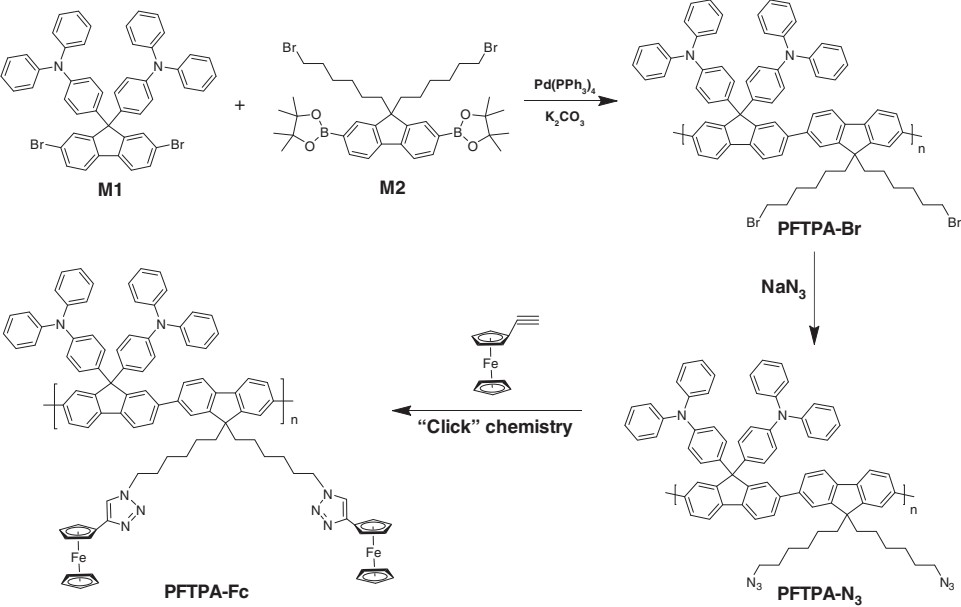

**Fig. 1** Synthesis of the redox active polymer PFTPA-Fc

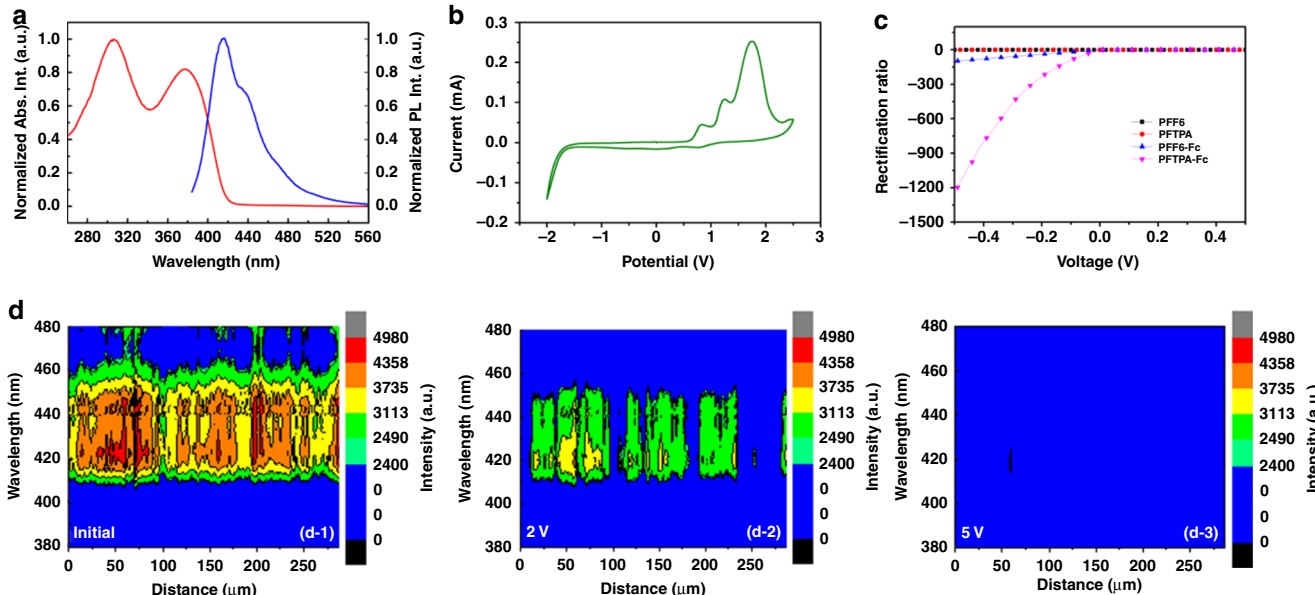

**Fig. 2** Spectroscopic and electrical characteristics of PFTPA-Fc. **a** UV-Visible absorption (red) and fluorescence (blue) spectra and **b** cyclic voltammetry plot of PFTPA-Fc. **c** Current–voltage of the ITO/polymer/Pt sandwich structures device constructing from different polymers of PFF6, PFTPA, PFF6-Fc, and PFTPA-Fc, respectively. **d** Fluorescence line scan image of the ITO/PFTPA-Fc/Pt device at different oxidation state, each composed of 122 spectra acquired along a 287 μm straight line

π–π* electronic transition of the conjugated polyfluorene backbone, while the longer wavelength peak with lower intensity is due to the vibrionic coupling between the n–π* and π–π* electronic transitions of the ferrocene chromophore[37–39]. The fluorescence spectra of PFTPA-Fc shows sharp emission at about 415 nm, as well as moderate and broad shoulders at 438 and 472 nm, respectively. The emission peak at 415 nm can be attributed to the monomer fluorescence of the polymer backbone, whereas the broader shoulder at 472 nm is assigned to the π–π stacking of the conjugated backbones. In comparison to the optical spectra of the control samples containing no or only one

of the TPA or Fc pendant groups (Supplementary Note 4, Supplementary Figure 7), the appearance of the moderate emission at 438 nm in PFTPA-Fc confirms the successful linking of ferrocene moieties onto the polymer.

**Electrochemical and electronic characteristics of PFTPA-Fc.** Cyclic voltammetry of PFTPA-Fc reveals triple oxidation behavior with the onset oxidation potentials of 0.61, 1.08, and 1.41 V, respectively, which is in good agreement with the presence of ferrocene, triphenylamine and fluorene moieties in the polymer

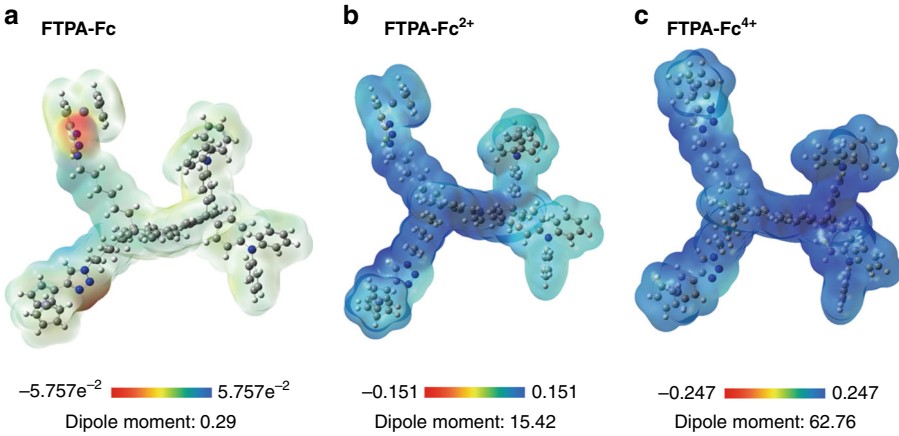

**Fig. 3** Electrostatic potential (ESP) distributions of the FTPA-Fc repeating unit. Simulated ESP profile in different oxidation states of **a** prinstine FTPA-Fc, **b** FTPA-Fc$^{2+}$, and **c** FTPA-Fc$^{4+}$, respectively. The gray, white, blue and light purple spheres represent carbon, hydrogen, nitrogen, and iron atoms, respectively

(Fig. 2b). As shown in Supplementary Figure 8, PFF6 containing none of the TPA or Fc pendant groups shows oxidation at the onset potential of 1.68 V, while PFTPA or PFF6-Fc with only one of the triphenylamine or ferrocene pendant moieties show additional oxidation peaks at the onset potentials of 0.95 and 0.72 V. These results indicate that the triple oxidation behavior of PFTPA-Fc with the onset potentials of 0.61, 1.08, and 1.41 V can be assigned to that of the ferrocene and triphenylamine pendant groups and the polyfluorene backbone, respectively. It is noteworthy that the introduction of Fc moiety onto the polymer shifts the onset oxidation potential of the fluorene backbone to lower value (for instance, 1.32 V in PFF6-Fc and 1.41 V in PFTPA-Fc) obviously, suggesting that the electron-rich ferrocene groups influence the electronic structure of the polymer materials significantly. The onset oxidation potential of the polyfluorene backbone in PFTPA remains ~1.70 V. Consistent with the electron-rich nature of either the fluorene backbone and the TPA/Fc pendant unities, obvious reduction peaks are not observed in the as-synthesized p-type semiconducting copolymer of PFTPA-Fc. The copolymer carries a moderate bandgap of ~2.96 eV as derived from the absorption edge of the UV-Visible spectra, with the highest occupied molecular orbital (HOMO) and the lowest unoccupied molecular orbital (LUMO) energy levels of −5.03 and −2.07 eV that are estimated from the electrochemical results, respectively[36]. The HOMO level is further verified through ultraviolet photoelectron spectroscopic (UPS) measurement[40,41], which shows a similar result as shown in Supplementary Figure 9. The high HOMO of PFTPA-Fc makes it an efficient hole transporting material when sandwiched between a pair of indium-tin oxide (ITO) and platinum (Pt), showing remarkable self-rectifying behavior that can is of great importance for accurate reading of the designated cell in large-scale memory arrays (Fig. 2c)[42]. In this work, the voltages are all applied to the Pt bottom electrodes. The high thermal degradation temperature of 354 °C also endows PFTPA-Fc good thermal stability for device applications (Supplementary Figure 10). The HOMO levels of PFF6-Fc, PFTPA, and PFF6 were lower from −5.09 and −5.37 to −6.10 eV, respectively (Supplementary Note 5), which in turn depress the hole-transporting capability of the material and lead to sequentially decreasing self-rectifying characteristics of the respective ITO/polymer/Pt device.

The solid-state electrochemical redox phenomena of PFTPA-Fc thin film is further probed via fluorescence measurements. The use of transparent conductive indium-tin oxide as the top electrodes allows in situ optical properties monitoring of the device. To investigate the spatial evolution of the oxidative states of the PFTPA-Fc thin film, fluorescence line scans were conducted by acquiring ~122 spectra along equally spaced points in a 287 μm straight line on top of the ITO/polymer/Pt device (Supplementary Figure 11). Figure 2d shows the change of the sample's fluorescence intensity as false color under different applied voltages, with its wavelength plotted along the Y axis and the sampling position depicted along the straight line on the X axis. In accordance with Fig. 2a, the as-prepared device exhibits strong emission in the wavelength range of 410–460 nm (left panel of Fig. 2d). Moderate emissions between 460 and 480 nm associated with the π–π stacking of the conjugated backbones are also observed non-uniformly along the sampling line, indicating that the PFTPA-Fc film fabricated by spin-coating only crystallizes in minority, if crystallization exists. The fluorescence signals centered between 430 and 460 nm attenuates relatively more significantly when the device is subjected to 2 V voltage (middle panel of Fig. 2d, with ITO electrode positively biased), suggesting that the ferrocene (Fe$^{2+}$) moieties are partially oxidized into ferrocenium (Fe$^{3+}$) forms. The conjugated polymer backbone and TPA pendant emission between 415 and 430 nm also weakens under voltage stress, although the quenching efficiency are relatively lower. Interestingly, the changes in the fluorescence characteristics of the polymer film are highly localized in space, and the positions with initial low emission intensity (e.g., at around 100, 175–190, and 225–280 μm) experience more obvious fluorescence quenching. This may be arising from the local rough polymer/electrode interface and non-uniform distributed electric field inside the PFTPA-Fc thin film, and is consistent with the filamentary switching nature of the well explored memristor devices[18,43,44]. When 5 V voltage is applied to the device to oxidize all the pendant ferrocene/TPA groups and the polymer backbone, the fluorescence signal of the polymer layer vanish completely (right panel of Fig. 2d). In the second oxidation stage under higher voltages, the majority parts of PFTPA-Fc that are oxidized are the conjugated polymer backbone and triphenylamine moieties, as the quenching of fluorescence spectra occurs more significantly in the wavelength range between 415 and 430 nm. The electrochemical behavior of the solid-state PFTPA-Fc device well follows its multilevel redox properties as observed in liquid electrolyte during cyclic voltammetry measurements.

To assess the redox-related charge carrier transport properties of PFTPA-Fc, static electronic scenario of the repeating unit of

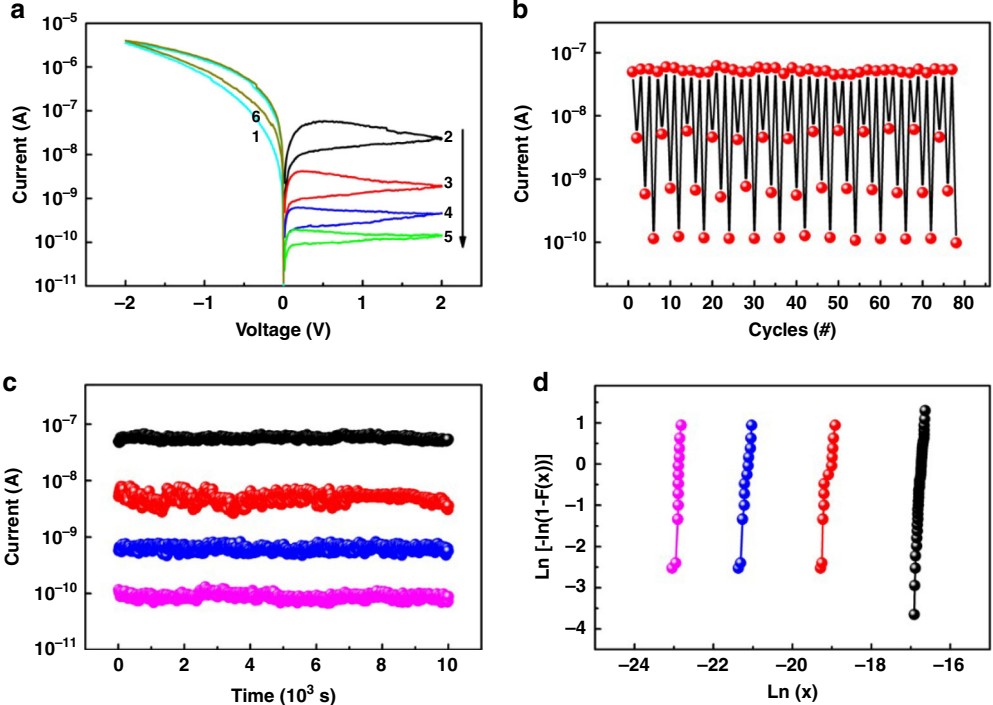

**Fig. 4** Multilevel memristive switching characteristics in high resistance state. **a** Current–voltage curves of the ITO/PFTPA-Fc/Pt device. **b** Endurance and **c** retention performance of the multilevel storage. **d** Weibull distribution of the device current in the different conductance stages

the copolymer is investigated through molecular simulation using Gaussian program package and the density functional theory (DFT)[45,46]. As shown in Fig. 3a, the ground state repeating unit containing fluorine backbone with pendant triphenylamine and ferrocene groups (denoted as FTPA-Fc) shows positive electrostatic potential (ESP) channel in light blue color throughout the entire molecule, with the nitrogen atoms of the triazole bridging group bearing negative ESP spots (yellow) and serving as potential charge traps in the polymer thin films. The energy bandgap (lowest unoccupied molecular orbital (LUMO)–highest occupied molecular orbital (HOMO)) and dipole moment of the neutral ground state FTPA-Fc unit are 3.54 eV and 0.29 Debye, respectively (Supplementary Figure 12a). When the ferrocene unities of the polymer are first oxidized under low voltage, the negative ESP traps of the triazole groups disappear while the electrostatic potential of the FTPA-Fc unit becomes more positive with darker blue color (Fig. 3b). Oxidation can significantly decrease the energy bandgap of the polymer, which lowers to 1.15 eV in the FTPA-Fc$^{2+}$ molecule and will make the polymer thin film more conductive with relative ease of charge carrier transition across the bandgap (Supplementary Figure 12b). The charged repeating unit also exhibits a larger dipole moment of 15.42 Debye. It should be pointed out that during molecular simulation we didn't introduce any counter ion to balance the charge of the oxidized FTPA-Fc$^{2+}$ molecule in the vacuum environment, thus the absolute values of the HOMO and LUMO energy levels would be different from that of the actual values. Nevertheless, the lowered energy bandgap of PFTPA-Fc will result in device transition into a relatively lower resistance state. Further oxidation of the triphenylamine groups makes the electrostatic potential of the FTPA-Fc$^{4+}$ molecule even more positive with a greater dipole moment of 62.76 Debye (Fig. 3c). The much lower energy bandgap of 0.15 eV will consequently turn the polymer device into metallically conductive state (Supplementary Figure 12c).

**Memristive data storage and arithmetic features of PFTPA-Fc**. In accordance with the consecutive oxidation characteristics of PFTPA-Fc in solid thin film forms, the ITO/PFTPA-Fc/Pt devices show interesting memristive switching behaviors in either the high resistance state (HRS) or low resistance state (LRS). HRS currents monitored between ±2 V show four distinguishable stages that are useful for the realization of multilevel memory (Fig. 4). LRS currents exhibit consecutive modulation and can be utilized in information processing applications (Fig. 5). The low-voltage memristive behavior in HRS and LRS are independent yet inter-convertible, wherein an abrupt bistable resistive switching operation at high-voltage level (± 4 V) can reconfigure the device from one mode to another (Supplementary Figure 13). Referring to the oxidation behavior of PFTPA-Fc as shown in Figs. 2 and 3, the underlying mechanism of its unique device characteristics can be understood as following. The HOMO/LUMO energy levels and electrode work functions alignment suggests that hole injection from ITO into the PFTPA-Fc layer in the negatively-biased voltage sweep is a favored process[34,36,47–50]. When the electric field is low, charge carriers do not possess sufficient energy to overcome the Schottky barrier at the ITO/PFTPA-Fc interface, and the small current as observed is likely due to tunneling of the charge carriers. When the applied voltage exceeds the Schottky barrier height of ~0.23 eV, holes are injected into the polymer and can migrate through the continuous positive ESP channel of the polymer backbone (in light blue color of Fig. 3a). When the applied voltage reaches the oxidation potential of the ferrocene moiety in the low-voltage range of 0 to ± 2 V, the pendant Fc unities become oxidized and give rise to the four-stage switching in HRS of the polymer. Ramping of the applied voltage to −4 V can initiate the subsequent fast oxidation of the triphenylamine chromophore, giving rise to the abrupt current jump observed at −3.3 V and programming the device from HRS to LRS (Supplementary Figure 13). In LRS the redox

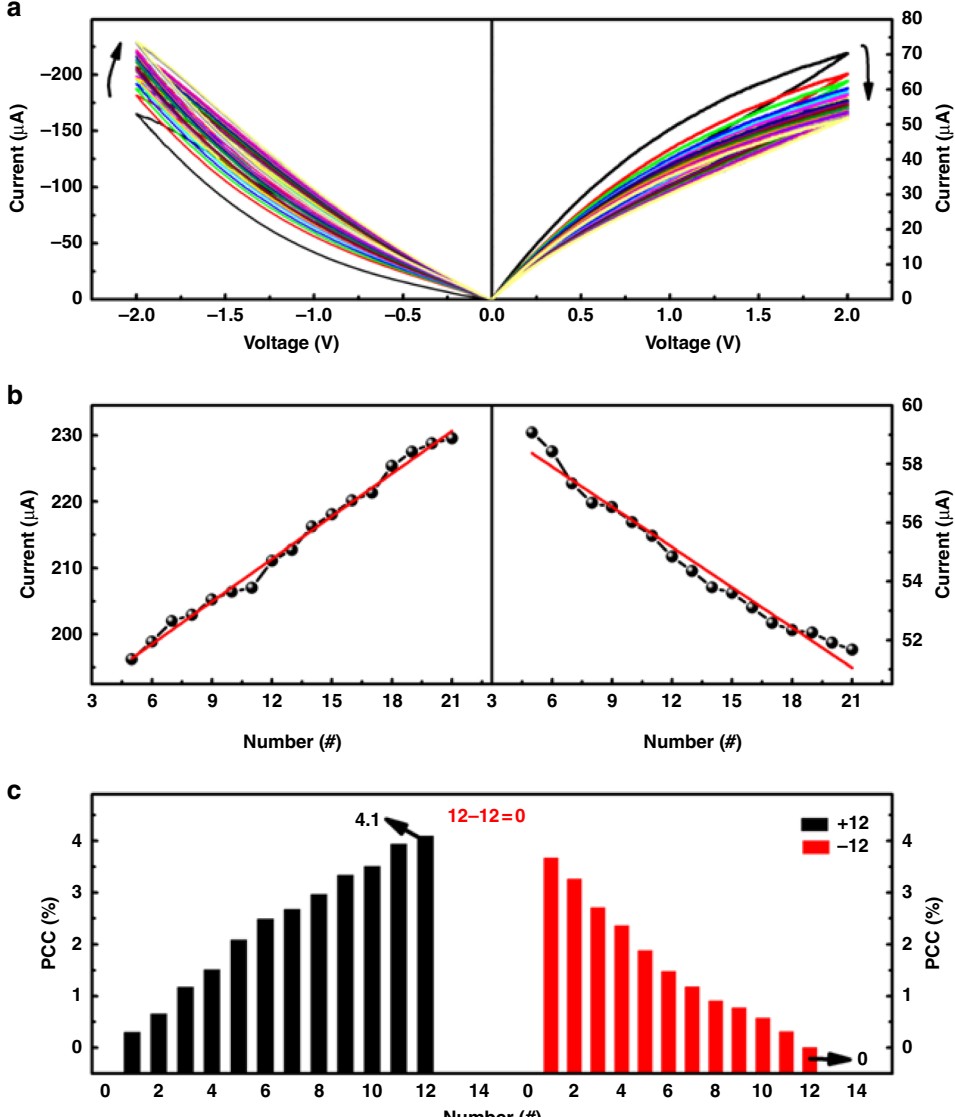

**Fig. 5** Consecutive memristive switching characteristics in low resistance state. **a** Current–voltage curves of the ITO/PFTPA-Fc/Pt device. **b** Linear relationship between the device currents read at ± 2 V and voltage sweeping numbers. **c** Calibration of the PFTPA-Fc memristor with the operation of 12–12 = 0 for decimal arithmetic calculations

behavior of the TPA groups is ceased when being scanned between ± 2 V, and the continuous modulation of device current is ascribed to the Fc redox behavior again (Fig. 5a). PFF6-Fc with only Fc pendants also exhibits multilevel switching behavior (Supplementary Figure 14a), while abrupt bistable resistive switching characteristics are observed in PFTPA that contains only TPA chromophores (Supplementary Figure 14b). In comparison, PFF6 with none of the redox active groups shows insulating nature (Supplementary Figure 14c). The current–voltage curves of these control samples again confirms that the low-voltage level consecutive switching and high-voltage level bistable switching behaviors of PFTPA-Fc are attributed to the oxidation of ferrocene and triphenylamine groups, respectively. Being different from those documented in the literature using bistable switching behavior of ferrocene containing materials for binary data storage applications[27,33,34,51–56], the continuous modulation of the material's conductance in a single voltage polarity may

make the integration of multilevel memory (with enhanced storage density) and computing capability into a single memristor based processing-memory unit possible through redox induced memristive switching in a natural accumulation manner.

Instead of storing 1 bit information per cell binarily, the PFTPA-Fc device demonstrates multilevel memory capability in its HRS state. Initially, the ITO/PFTPA-Fc/Pt device shows low current and a rectifying feature (Fig. 2c). Application of the scanning voltage in the direction of 0 V to −2 V to 0 V (sweep 1, with ITO top electrode positively biased) can oxidize the ferrocene ($Fe^{2+}$) moieties to ferrocenium ($Fe^{3+}$) form, and turns the device into a relatively higher conductive stage (Fig. 4a). This serves an "initialization" step for electronic devices. Subsequent reversed scannings in the positive direction can reduce the ferrocenium back to ferrocene gradually, as four consecutive sweeps (2–5) between 0 and 2 V decrease the device current to a low level. The programing of current to different levels can be

viewed as the writing of multiple bits of information into the device. It is noteworthy that the immediately following negative sweep 6 almost repeats the I–V footprint of sweep 1, indicating that the four positive voltage scannings have reduced ferrocenium to ferrocene completely and finish the "write–read–erase" operating loop. Therefore, the stepwise current modulation behavior, together with the stability of the oxidized ferrocenium chromophores, gives PFTPA-Fc device the possibility of non-volatile multilevel information storage. Four-level memory performances are further evaluated in pulse operation mode (Fig. 4b), wherein the device can be switched between different stages repeatedly. The ITO/PFTPA-Fc/Pt device is first programmed to a relatively higher conductive stage with a negative voltage pulse of −2 V amplitude and 30 ms width (read at 0.2 V), and then written to different storing stages by applying multiple positive pulses of 2 V and 30 ms. All the four stages are stable under the 0.2 V constant reading voltage for $10^4$ s without any obvious degradation (Fig. 4c), and their conductances are distributed in very narrow ranges as verified through Weibull analysis (see Fig. 4d and Supplementary Note 6). The good retention and endurance characteristics of the ITO/PFTPA-Fc/Pt memristor, which shows promising uniformity of the switching voltages and device resistances, are of great importance for practical applications. Noting that the today's state-of-art CMOS circuits are capable of accurately distinguishing different resistance states with the ON/OFF ratio of ~10, multilevel storage in the present polymer device with one order of difference between the resistances of the four states may not lead to obvious misreading issues[57,58]. For the in-memory computation applications as discussed below, small ON/OFF ratio can nevertheless ensure larger numbers of possible resistance states of the device, which are crucial for allowing highly efficient neuromorphic computing tasks[9,59,60].

Application of a negative scan between 0 and −4 V serves as the reconfiguring operation that transforms the PFTPA-Fc device from HRS memory to LRS processor mode (Supplementary Figure 13). Generally, the decimal arithmetic function is developed from counting quantities, which is also valuable for designing new concept computation beyond the binary system with a higher-logic paradigm. In the present PFTPA-Fc device, the application of negative voltages that increases the device current equals the addition function of an abacus, while the application of positive voltages is used to make subtraction operations. The quantity of the input voltage stimuli is counted by monitoring the percent changes of the device current (PCC) with respective to the initial value. To function normally as a decimal abacus, a good linear relationship between the numbers of the applied voltage stimuli and the device current is a basic requirement. As shown in Fig. 5a, the device current can be modulated continuously when being subjected to consecutive negative (0 V to −2 V to 0 V) and positive (0 V to 2 V to 0 V) voltage sweeps. The relationship between the device currents read at ± 2 V and sweeping numbers is approximately linear (Fig. 5b). For calibration an $A–A = 0$ operation is conducted in pulse mode to guarantee the abacus accuracy (Fig. 5c). The change in device current increases to 4.1% after the succession of 12 consecutive negative pulses (−1 V and 30 ms). As such, the number 12 will be counted in future calculation when a PCC of 4.1% is read. By defining the natural number 0 as a PCC of 0%, the decimal numbers of 0–12 can be indexed proportionally. The PCC returns to 0% when a subsequent train of 12 consecutive positive pulses (2 V and 30 ms) is executed, which realizes the subtraction operation of 12–12 = 0 and calibrates the PFTPA-Fc memristor for accurate decimal arithmetic operations.

Commutative addition, subtraction and multiplication, as well as fractional division are implemented with the PFTPA-Fc memristive abacus. As shown in Fig. 6a, the application of a train of seven consecutive negative pluses (−1 V and 30 ms) followed by an extra series of five pulses with the same amplitude and duration gives a PCC of 4.1%, which exactly represents the number 12 and confirms 7 + 5 = 12. Reversing the loading order of the two sets of input signals again results in the PCC of 4.1%. All the pulses are the same herein and thus verifies the commutative law that $A + B = B + A$ (7 + 5 = 5 + 7 = 12). Figure 6b demonstrates the commutative subtraction operation. After the loading of 12 negative pulses of −1 V and 30 ms which generates the PCC of 4.1%, the sequential application of a train of seven positive pulses of 2 V and 30 ms and a following train of five same positive pulses reduces the PCC to 0%. The superposition of two sets of five positive and seven positive pulses in that order onto the preloaded 12 negative pulses also reduces the PCC to 0%, which confirms that $A–B–C = A–C–B$ (12–7–5 = 12–5–7 = 0). Similarly, multiplication that is realized based on the accumulative addition operation also obeys the commutative law (Fig. 6c). The net effect of three sets of four negative pulses of −1 V and 10 ms equals to that of another four sets of the three same negative pulses, assuring that $A × B = B × A$ (4 × 3 = 3 × 4 = 12). Finally, the fractional division function is completed by combining the subtraction and addition operations according to the following philosophy. For $A ÷ B$ the successive subtraction of "A–B–B…" will continue unless the remainder (r) is smaller than the divisor B. The quotient is counted as the numbers of the performed subtraction operations. Afterward the replacement of r by $r × 10$ (or, $r + r × 9$) is carried out to continue the successive subtraction. The subtraction-replace-subtraction until the remainder reaches 0, and the division completes with the final quotient counted as the numbers of subtraction operations performed. Here we take 7 ÷ 5 = 1.4 as an example with our PFTPA-Fc memristor (Fig. 6d). The device is first reset to 0 for initialization. Then a series of seven negative pluses of −1 V and 30 ms is loaded, followed by another train of five positive pluses of 2 V and 30 ms, to perform 7–5. An integer quotient of 1 and a remainder of 2 are obtained. Since the remainder 2 is smaller than the divisor 5, replacement of 2 by 2 × 10 (2 + 2 × 9) is done by adding two trains of nine positive pulses to the device. Subsequently repeated subtraction of −5 for four times (20–5–5–5–5) results in the final PCC of 0% and terminates the fractional division calculation. The only one time replacement operation performed during calculation indicates that the decimal number of 4 is at tenths position and the quotient of 7 ÷ 5 is 1.4 (1 + 0.4).

**Binary Boolean logic computing with PFTPA-Fc**. Boolean logic computing is another pivotal part for information processing, which can be enable by the accumulative resistive switching characteristics of the PFTPA-Fc memristor. For demonstration we perform the OR gate execution in the LRS state. As shown in Fig. 7a, the voltages applied to the top and bottom electrodes are defined as input A and input B, respectively; whereas the positive pulse with an amplitude of 2 V and duration of 30 ms and negative pulse of −2 V and 30 ms are defined as input logic values 0 and 1. The initial device current read at 0.2 V is 10.36 µA, and is used as a criterion to differentiate the output signals. If the device current at 0.2 V is higher than 10.36 µA after certain operations, then the output logic value is 1. Otherwise the result is 0. Figure 7b shows the truth table and experimental results of the device current in response to different combinations of input pulses. It is clear that the application of input logic 1 to either the top or the bottom electrodes, or both, can change the device

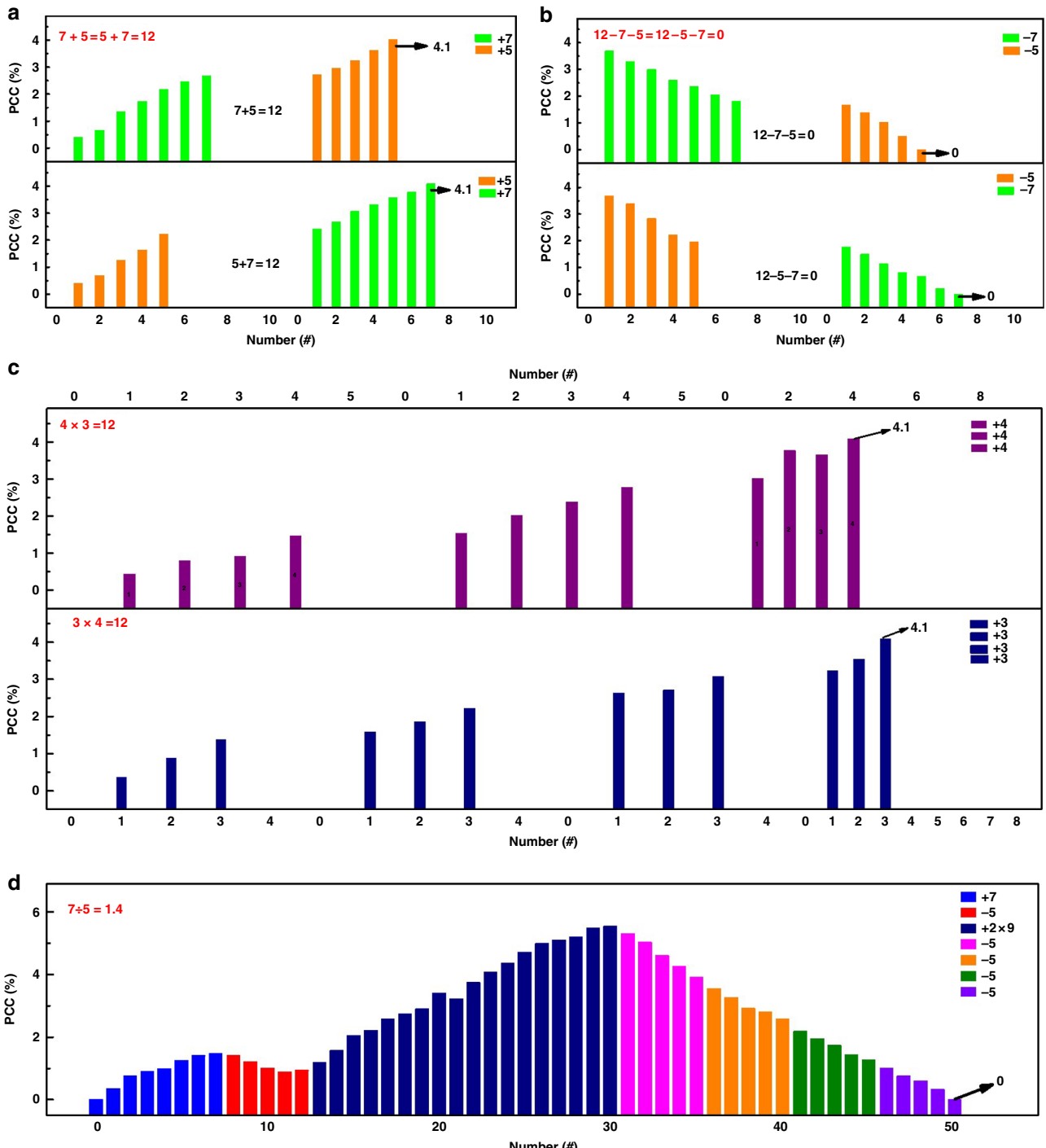

**Fig. 6** Demonstration of arithmetic computing with the PFTPA-Fc memristor. **a** Commutative addition, **b** subtraction, and **c** multiplication, as well as **d** fractional division conducted with the ITO/PFTPA-Fc/Pt device

current to higher than 10.36 µA and gives the logic output of 1. On the other hand, the application of input logic 0 to both electrodes makes the device current remaining lower than 0.36 µA and leads to the logic output 0. This well resembles the OR gate operation of the Boolean logic algorithm, and can be further extended to other logic operations with proper operating philosophy and auxiliary circuit designs.

## Discussion

To summarize, we demonstrate a proof-of-concept polymer memristive processing-memory unit that integrates programmable multilevel storage, decimal abacus and logic functionalities. By employing the solid-state electrochemical redox reaction of the pendant ferrocene and triphenylamine moieties, the PFTPA-Fc based device exhibits independent and inter-convertible four-level information storage and all of the four basic arithmetic functions. It should be noted that since the electrochemical redox characteristics of the polymer is highly dependent on its chemical structure and composition, variation in the amount of ferrocene and/or triphenylamine moieties tethered onto the pendant of PFTPA-Fc will surely leads to different resistive switching characteristics. The film thickness of the

**a**

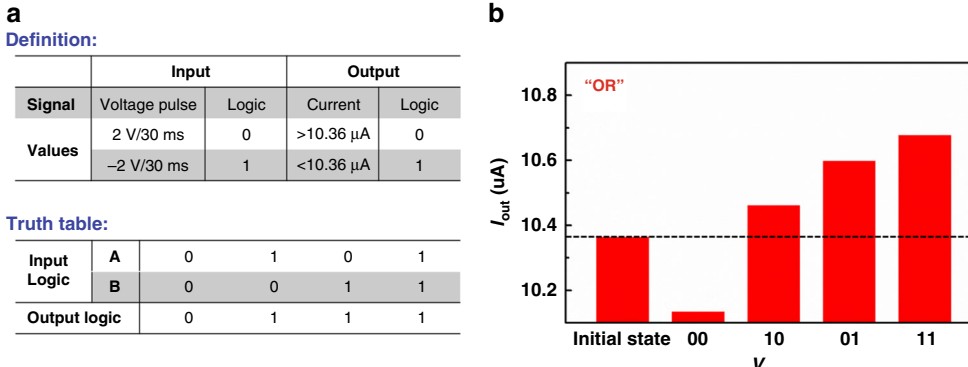

**b**

**Fig. 7** Realization of the OR logic gate function with PFTPA-Fc memristor. **a** Definition for the logic inputs/outputs and truth table. **b** Experimental results of the PFTPA-Fc device

polymer may also influence the device performance[61–65], e.g., HRS state resistance, ON/OFF ratio and etc. The structure–property relationship between the Fc and TPA group content (and their spatial arrangement in the polymer), the polymer film thickness and the device performance thus require extensive investigation, which should be explored systematically in the future as a supplement to the present study on the conceptual demonstration of polymer memristor device. Nevertheless, compromising the processor's feature with Boolean logic operation, the present polymer memristive approach will provide insights into the development of high-performance intelligent computing beyond the conventional von Neumann paradigm.

## Methods

**Synthesis and characterization**. The details for the synthesis of the polymers are summarized in the Supplementary information. Nuclear magnetic resonance (NMR) spectra were recorded on a Bruker 400 spectrometer at a resonance frequency of 400 MHz for $^1$H and 100 MHz for $^{13}$C in deuterated solution with a tetramethylsilane (TMS) as a reference for the chemical shifts. Molecular weights were determined with a Waters 2690 gel permeation chromatography (GPC) using a polystyrene standards eluting with tetrahydrofuran. Fourier transform infrared (FTIR) spectra were performed on a Nicolet Nagma-IR 550 spectrophotometer using KBr pellets. X-ray photoelectron spectroscopy (XPS) measurements were carried out on an ESCALAB 250Xi (Thermo Fisher) with Al Kα radiation as X-ray source for radiation. Ultraviolet photoelectron spectroscopic (UPS) measurement was performed on Kratos Axis Supra. Thermogravimetric analyses (TGA) were carried out using Pyris 1 TGA. The UV-Visible absorption spectral measurements were carried out with a Shimadzu UV-2450 spectrophotometer. A HORIBA JOBIN YVON Fluoromax-4 spectrofluorometer was used to record the steady-state fluorescence spectra. Both the UV-Visible absorption and fluorescence spectra of the polymers samples were measured under the same condition in diluted toluene solutions. The organic solutions were excited at the wavelength of 365 nm for fluorescence measurement. Cyclic voltammetry (CV) measurements were performed on a model CHI 650D electrochemical workstation using $n$-Bu$_4$NClO$_4$ (0.10 M) in acetonitrile as the supporting electrolyte, a platinum disk as working electrode, an Ag/AgCl electrode as reference, and a Pt wire as counter electrode. Dry HPLC grade acetonitrile was degassed with high purity Argon to eliminate possible interference arising from the environmental moisture and oxygen species. In situ fluorescence studies of the solid-state polymer thin film were also carried out at room temperature using a Renishaw Via-reflex spectrometer with Keithley 237 voltmeter. The excitation was provided by a He–Ne laser ($\lambda_{exc} = 325$ nm). The spectral resolution was better than 1 cm$^{-1}$.

**Molecular simulation**. All density functional theory (DFT) and time-dependent DFT (TD-DFT) calculations were performed using the Gaussian 09 package[43]. For better comparison between different charged compounds, all states for geometry optimization and excited-state calculation are in singlet and close shell. The B3LYP functional was used for geometry optimization in the ground state. The 6-31G(d) basis set was used for the C, H, and N atoms, while LANL2DZ and its corresponding pseudopotential was used for Fe atoms. All geometry optimization was done in the gas phase.

**Device fabrication and characterization**. The memristive switching behaviors of the polymer materials were examined in ITO/polymer/Pt structures. The Pt/Ti/

SiO$_2$/Si substrates (Hefei Ke Jing Materials technology Co., LTD.) were pre-cleaned in the ethanol, acetone and isopropanol in an ultrasonic bath, each for 30 min in that order. The thicknesses of the Pt and Ti layers are 150 and 20 nm, respectively. Polymer solution of 10 mg/mL was prepared by dissolving the polymer powders in toluene. The as-prepared solutions were filtrated through polytetrafluoroethylene (PTFE) membrane micro-filters with a pore size of 0.45 μm to remove any dissolved particles. The polymer layers were then deposited by spin-casting 50 μL solution of the polymer onto the pre-cleaned Pt/Ti/SiO$_2$/Si substrate at a spinning speed of 600 rpm for 15 s and then at 1000 rpm for 50 s, followed by being vacuum-dried at 50 °C overnight. The thickness of the polymer layers are ~130 nm. The ITO electrodes with the thickness of 100 nm and diameter of 100 μm were deposited by sputtering an ITO target using pulsed laser deposition (PLD) technique (PLD300 system from SKY Technology CAS, equipped with COMPEXPRO 205 excimer laser generator from Lambda Physik) and patterned through a metal shadow mask. The electrical properties of the ITO/polymer/Pt devices were measured with a Keithley 4200 semiconductor characterization system under s sweep or pulse mode at room temperature.

## Data availability

The authors declare that the main data supporting the findings of this study are available within the article and its Supplementary Information files. Extra data are available from the corresponding author upon request.

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

## Acknowledgements

The authors acknowledge the financial supports from the National Natural Science Foundation of China (51333002, 21404037, 61722407 and 61674153), the National Key R&D Program of China (2017YFB0405604), the Natural Science Foundation of Zhejiang Province (LR17E020001), Key Laboratory of Nanodevices and Applications, Suzhou Institute of Nano-Tech and Nano-Bionics, Chinese Academy of Sciences (18CS01), the Chenguang Program (15CG28), and the Pujiang Program (16PJ1402400).

## Author contributions

B.Z., F.F., G.L., and Y.C. conceived the idea. B.Z. and F.F. synthesized and characterized the polymers. F.F. and W.H.X. conducted the electrical measurements. Y.B.F. and X.D.Z. performed the molecular simulation. B.Z., F.F, W.H.X., G.L., Y.B.F., X.-H.X., J.G., R.-W.L., and Y.C. co-wrote the paper. All the authors discussed the results and commented on the manuscript.

## Additional information

**Competing interests:** The authors declare no competing interests.

