## [Peer Review File · Nature Communications]

Reviewers' comments:

Reviewer #1 (Remarks to the Author):

This manuscript prepared by Chen, Li and Liu was mainly focused on the preparation of a polymer based memristor device. First they synthesized a conjugated polymer pendent with f triphenylamine (TPA) and ferrocene (Fc), then a memristor device, ITO/PFTPA-Fc/Pt, was prepared and exhibited multilevel memory behavior and four basic decimal arithmetic operations. This result is undoubtedly exciting because it may be the first example for polymer device to demonstrate the real information storage. This result also encouraged other groups to extend the application of organic memory devices.

However, before the acceptance of this manuscript, some revisions are necessary.

The suggestions are listed as follows.

- (1) The authors should provide CNMR of monomers and polymer.
- (2) How many Fc moieties were pendent on the polymer backbone?
- (3) In Figure S4, why does the pair of redox potentials of Fc disappear? It is strange. In addition, the onset oxidation potential at 1.08V may belong to that of water. Please check. I strongly suggest they do the experiment again.
- (4) Fluorescence spectra (Figure S3) of three polymers should be measured under the same condition, The authors didn't list the experimental condition. Are they at solid film or in the organic solution?
- (5) In line 136, the authors mentioned 'a small bandgap of ~ 2.96 eV'. I can't agree this value means a small band gap.
- (6) In line 238, I want to know how long they wait when the last sweep finished? How about the device performance when the interval time changes (longer or shorter)?

Reviewer #2 (Remarks to the Author):

In this manuscript, authors reported a new ferrocene-containing polymer (PFTPA-Fc) with multilevel electrical stability. The electrical properties of the as-synthesized polymer were investigated both experimentally and theoretically. The results of sandwich-like diode device indicate that PFTPA-Fc is suitable for multilevel memory due to its different conductivities at high and low voltages. Moreover, authors also explored the practical application of this material by integration of its multilevel memory and computing capability into a single memristor device. The manuscript was generally well organized and well written. However, I do not think that it is suitable for publication in such high-quality journal but more suitable for journals such as Macromolecules or Polymer Chemistry etc. Below are some comments for authors' attention.

- (1) Have authors confirmed the quantity of the grafted Fc moieties? How is the relationship like between the quantity of grafted Fc moieties and electrical property of the polymer?
- (2) How about the solubility of PFTPA-Fc? Can it be fully dissolved in toluene or other organic solvents? Since the structure of the material can significantly influence the device performance, so these two concerns should be addressed carefully.
- (3) What's the size of the final memristor device?
- (4) Since the TFT structure can be well compatible in the industrial ICs, why authors not consider the transistor device for this material?
- (5) There are already some works using Fc in developing polymers for various organic memory applications. How are they different from yours in terms of performance or what are the advantages of these new devices?

Reviewer #3 (Remarks to the Author):

This manuscript contains significant contribution to unconventional memristor-based computing . The results reported are impressive and have great potentials. I recommend prompt publication in view of its timeliness.

Reviewer 1#:

This manuscript prepared by Chen, Li and Liu was mainly focused on the preparation of a polymer based memristor device. First they synthesized a conjugated polymer pendent with f triphenylamine (TPA) and ferrocene (Fc), then a memristor device, ITO/PFTPA-Fc/Pt, was prepared and exhibited multilevel memory behavior and four basic decimal arithmetic operations. **This result is undoubtedly exciting because it may be the first example for polymer device to demonstrate the real information storage. This result also encouraged other groups to extend the application of organic memory devices.** However, before the acceptance of this manuscript, some revisions are necessary. The suggestions are listed as follows.

Answer: We thank the reviewer for the very positive comment on our work. We have carefully considered all the reviewers' comments and revised the manuscript accordingly as described below.

Question 1: The authors should provide CNMR of monomers and polymer.

Answer: Thanks. We have included the CNMR data for monomers and polymers in the revised Supplementary Information on page 15-19. Together with the FTIR and

XPS results, it can be confirmed that all the targeted materials are obtained successfully.

Question 2: How many Fc moieties were pendent on the polymer backbone?

Answer: Thanks. According to the FTIR and NMR data, each repeating unit of the PFTPA-Fc polymer contains two ferrocene unities, as shown in **Figure 1**. Related discussion is included in the revised manuscript on page 6 lines 15-18 and in the revised supplementary information on page 17 lines 11-20.

Question 3: In Figure S4, why does the pair of redox potentials of Fc disappear? It is strange. In addition, the onset oxidation potential at 1.08V may belong to that of water. Please check. I strongly suggest they do the experiment again.

Answer: Thanks. We have carefully re-done the cyclic voltammetry measurements and confirm that PFF6-Fc does show the pair of redox potentials of the ferrocene moiety. A new cyclic voltammetry spectrum showing clearer reversible redox behavior of PFF6-Fc is plotted in the Supplementary Figure 4 in the revised SI file. For a better illustration, the cyclic voltammetry spectrum obtained in the positive potential branch is also shown below. Without including the strong oxidative peak of the polyfluorene backbone and the intensive tail in the negative voltage range, the pair of reversible ferrocene redox peaks can be seen clearly. On the other hand, during our experiment dry HPLC grade acetonitrile was degassed with high purity Argon to eliminate possible interference arising from the environmental moisture and oxygen species. As such, the onset redox potential at 1.08 V can be confidently ascribed to that of the triphenylamine pent groups. Related discussion is added in the revised manuscript in the Method section on page 18 line 20-22.

Question 4: Fluorescence spectra (Figure S3) of three polymers should be measured

under the same condition. The authors didn't list the experimental condition. Are they at solid film or in the organic solution?

Answer: Thanks. We thank the reviewer for his/her comment and suggestions. The fluorescence spectra of the three polymers plotted in Figure S3 were all measured under the same condition, e.g. monitored in diluted toluene solutions and excited at the wavelength of 365 nm. Related description on the experimental details is included in the revised manuscript on page 18 lines 14-17 as well as in the revised supplementary information on page 21 lines 2-4.

Question 5: In line 136, the authors mentioned 'a small bandgap of ~ 2.96 eV'. I can't agree this value means a small band gap.

Answer: Thanks. We have corrected the statement to "a moderate bandgap of ~ 2.96 eV" in the revised manuscript on page 8 line 3.

Question 6: In line 238, I want to know how long they wait when the last sweep finished? How about the device performance when the interval time changes (longer or shorter)?

Answer: Thanks. As shown in **Figure 4a**, the sweeps 2 to 6 were obtained immediately one after another, e.g. no waiting time was applied after any sweep. Since all the conductance states are stable for up to 10^4 s, applying waiting time with different intervals will not change device performance significantly.

Reviewer 2#:

In this manuscript, **authors reported a new ferrocene-containing polymer (PFTPA-Fc) with multilevel electrical stability.** The electrical properties of the as-synthesized polymer were investigated both experimentally and theoretically. The results of sandwich-like diode device indicate that PFTPA-Fc is suitable for multilevel memory due to its different conductivities at high and low voltages. Moreover, authors also explored the practical application of this material by integration of its multilevel memory and computing capability into a single memristor device. **The manuscript was generally well organized and well written.** However, I do not think that it is suitable for publication in such high-quality journal but more suitable for journals such as *Macromolecules* or *Polymer Chemistry* etc. Below are some comments for authors' attention.

Answer: We thank the reviewer for the comments on our work. We have carefully

considered your comments and revised this manuscript accordingly as described below.

Question 1: Have authors confirmed the quantity of the grafted Fc moieties? How is the relationship like between the quantity of grafted Fc moieties and electrical property of the polymer?

Answer: Thanks. During our work, we have used excess amount of ethynylferrocene to react with the azide moieties of the PFTPA-N₃ precursors through “Click” chemistry. As confirmed by the NMR and FTIR data, a 100% “Click” reaction has occurred to achieve complete grafting of ferrocene moieties onto the pendant of PFTPA-Fc, giving each repeating unit of the PFTPA-Fc polymers containing two Fc unities as shown in **Figure 1**. Related discussion is included in the revised manuscript on page 6 lines 15-18 and in the revised supplementary information on page 17 lines 11-20.

On the other hand, the quantity of grafted Fc moieties can certainly influence the electronic structure and electrical property of the polymer through electrochemical gating. However, the lowering of grafted Fc group quantity will inevitably leads to the incomplete consumption of the azide moieties during “Click” reaction and their involvement in the redox based resistive switching phenomena of the thin film device. This will complicate the electrical behavior and the underlying mechanism of the polymer device and therefore we did not try to explore the relationship between the quantity of grafted Fc moieties and electrical property of the polymer

Question 2: How about the solubility of PFTPA-Fc? Can it be fully dissolved in toluene or other organic solvents? Since the structure of the material can significantly influence the device performance, so these two concerns should be addressed carefully.

Answer: Thanks. PFTPA-Fc can be completely dissolved in toluene, THF or DMF to give a 10 mg/mL solution as shown in Supplementary Figure 3. Related discussion is included in the revised manuscript on page 6 lines 13-15.

Question 3: What's the size of the final memristor device?

Answer: Thanks. The PFTPA-Fc memristor devices explored in this study is round in shape and has a diameter of 100 μm. The thickness of the top ITO electrode and PFTPA-Fc switching layer are 100 nm and 130 nm, respectively. The thicknesses of the Pt and Ti layers of the commercial Pt/Ti/SiO₂/Si substrate are 150 nm and 20 nm, respectively.

Question 4: Since the TFT structure can be well compatible in the industrial ICs, why authors not consider the transistor device₄ for this material?

Answer: Thanks. Generally, the CMOS compatible transistor memories store data as the amount of charge trapped in the dielectric layer of the device. When the device thickness decreases to less than 10 nm, electrons can escape from the insulator easily. It results in spontaneous loss of the information stored in the device and deteriorates the reliability of the entire system significantly. This is known as the collapses of Moore's Law in the near future and efforts should be devoted to developing novel information techniques based on new structures and new mechanisms. In comparison, memristor stores information based on the conductance/resistance change of the device in response to an applied voltage or electric field. As such, the leaking problem of the conventional transistor based integrated circuits can be efficiently eliminated when the feature size of the devices decreases to less than 10 nm, which greatly improves the reliability of the memory chips. The two-terminal structure of the memristor also favors facile integration of the devices into crossbar array and three-dimensional stack for high density applications. With these concerns, we explored the memristor device with PFTPA-Fc. Of course, the material can be used to fabricate transistor device. Related discussion is included in the revised manuscript on page 3 line 19 to page 4 line 3.

Question 4: There are already some works using Fc in developing polymers for various organic memory applications. How are they different from yours in terms of performance or what are the advantages of these new devices?

Answer: As reported in the literatures (for instance, Mater. Design 2018-139-298; J. Polym. Sci. Polym. Chem. 2018-56-505; Dyes & Pigments 2017-146-210; JMCC 2016-4-921; Chem. Commun. 2015-51-13123; Chem. Commun. 2012-48-4235; Appl. Phys. Lett. 2011-98-232302; JACS 2007-19-9842; Synth. Metal 2007-157-640), redox active ferrocene materials with bistable resistive switching characteristics have been used for constructing binary memory devices. Each of these device can store 1 bit information. In our work, the consecutive modulation of the material's conductance in a single voltage polarity in the HRS and LRS states, achieved through redox induced memristive switching in a natural accumulation manner, makes the integration of multilevel memory (with enhanced data storage density) and computing capability into a single memristor based processing-memory unit (PMU) possible. Interestingly, the incorporation of TPA moiety results in an abrupt bistable resistive switching operation at high voltage level, which reconfigures the device between the multilevel storage and computing mode reversibly. As such, the new PMU device can provide a feasible solution to the forthcoming end of Moore's Law and the von Neumann bottleneck problem. Related discussion is included in the revised manuscript on page 11 line 13-

16 and on page 12 line 9-15. Related references are cited as Ref 27, 33, 34 and 45-50 in the revised manuscript.

Reviewer 3#:

This manuscript contains significant contribution to unconventional memristor-based computing. **The results reported are impressive and have great potentials. I recommend prompt publication in view of its timeliness.**

Answer: We thank the reviewer for the very positive comment on our work. We have carefully considered all the other reviewers' comments and revised the manuscript accordingly as described above.

Reviewers' comments:

Reviewer #1 (Remarks to the Author):

I'm satisfied with all revisions. And I'm pleased to recommend this manuscript accepted without further revision.

Reviewer #2 (Remarks to the Author):

Zhang et al. reported a new polymer PFTPA-Fc by reaction of triphenylamine (TPA) and ferrocene (Fc) onto the fluorine backbone through Suzuki coupling polymerization and click chemical reaction. The proof-of-concept memristive processing-memory unit based on PFTPA-Fc was explored and demonstrated for multilevel storage and logic operation. But there are still some key issues which are not clearly explained in this work.

i) The intrinsic conductive mechanisms of the device are not fully disclosed. How is the conductive mechanism between electrode and organic layer in the device?

ii) The on/off ratio is relatively low by comparing the state-of-art organic memory devices. The low on/off ratio will definitely influence the accuracy of resulting memory device.

iii) The energy level of the polymer thin film should be measured by UPS instead of just calculated by the spectra.

iv) Will the thickness of the polymer thin film influence the device performance? How about the grafting amount of the Fc and TPA? Will they also influence the device performance? How about the solubility of PFTPA-Fc in other organic solvents? Can some other environmental friendly solvents could be used to substitute toluene? These issues should be investigated carefully.

Therefore, I still think this work is not suitable for publication in Nat. Commun. in its current form but is more suitable for other polymer or materials science journals.

Prof. Dr. Yu Chen, Key Laboratory for Advanced Materials, School of Chemistry and Molecular Engineering, East China University of Science and Technology, 130 Meilong Road, Shanghai 200237, China

Professor Dr. Yu Chen
Alexander von Humboldt Research Fellow

Tel: +86-21-64253765
Fax: +86-21-64252485

E-mail: chentangyu@yahoo.com
yuchenavh@ecust.edu.cn

3. December 2018

We have prepared our new revised version in accordance with the comments raised by the second referee. The revised parts have been marked in red.

Specific response to the enquiries is given below.

Reviewer 2#:

Zhang et al. reported a new polymer PFTPA-Fc by reaction of triphenylamine (TPA) and ferrocene (Fc) onto the fluorine backbone through Suzuki coupling polymerization and click chemical reaction. The proof-of-concept memristive processing-memory unit based on PFTPA-Fc was explored and demonstrated for multilevel storage and logic operation. But there are still some key issues which are not clearly explained in this work.

Answer: We thank the reviewer for his/her comments on our work. We have carefully considered the reviewer's comments and revised the manuscript accordingly as described below.

Question 1: The intrinsic conductive mechanisms of the device are not fully disclosed. How is the conductive mechanism between electrode and organic layer in the device?

Answer: We thank the reviewer for his/her question. The conductive mechanism between the electrode and the organic layer is determined by the alignment of the electrode work functions and molecular orbital levels of the organic material. As revealed by the cyclic voltammetry and UV-Visible absorption spectroscopic measurements, the highest occupied molecular orbital (HOMO) and lowest unoccupied molecular orbital (LUMO) energy levels are -5.03 eV and -2.07 eV, respectively. With the electrode work functions of -4.8 eV and -5.6 eV for ITO and Pt electrodes, hole injection from ITO into the PFTPA-Fc layer in the negatively-biased voltage sweep is an energy favored process. When the electric field is low, charge carriers do not possess sufficient energy to overcome the Schottky barrier at the ITO/PFTPA-Fc interface, and the small current as observed is likely due to tunneling of the charge carriers. When the applied voltage exceeds the Schottky barrier height of ~ 0.23 eV, holes are injected into the polymer and can migrate through the continuous positive ESP channel of the polymer backbone (in light blue color of Figure 3a). As the applied voltage exceeds the oxidation of the Fc and TPA moieties, the pendant Fc and TPA units become oxidized and give rise to the unique resistive switching characteristics of PFTPA-Fc.

Related discussion on the interfacial charge transport mechanism is included in the revised manuscript on page 11 line 20 to page 12 line 8.

Question 2: The on/off ratio is relatively low by comparing the state-of-art organic memory devices. The low on/off ratio will definitely influence the accuracy of resulting memory device.

Answer: We thank the reviewer for his/her comments. Although the ON/OFF ratio of the present polymer device is relatively smaller than that reported in the literatures, the

today's state-of-the-art CMOS circuits are already capable of accurately distinguishing different resistance states with the ON/OFF ratio of ~ 10 . As such, the ON/OFF ratio with one order of difference between the resistances of the four states in the PFTPA-Fc memristor is no longer a critical issue that may lead to obvious misreading problem and was greatly concerned previously (*Rep. Prog. Phys.* 2012-75-076502; *Adv. Funct. Mater.* 2014-24-2171). In addition, for in-memory computation applications as discussed in the following section of the manuscript, small ON/OFF ratio can nevertheless ensure larger numbers of possible resistance states of the device, which are crucial for allowing highly efficient neuromorphic computing tasks (c.f., *Nature Electron.* 2018-1-22; *Nature Electronics* 2018-1-197, *Nature Electronics* 2018-1-386).

Related discussion on the ON/OFF ratio of the PFTPA-Fc device is included in the revised manuscript on page 14 line 6-12.

Question 3: The energy level of the polymer thin film should be measured by UPS instead of just calculated by the spectra.

Answer: We thank the reviewer for his/her suggestion. By using UPS measurement it is possible to obtain the HOMO level of the polymer. In order to verify the HOMO data that was measured by cyclic voltammetry measurement, we performed UPS measurement on PFTPA-Fc and the experimental result indicates that the HOMO level is ~ 4.90 eV (Supplementary Figure 6). The UPS data is similar to the CyV data of 5.03 eV, whereas the acceptable deviation can be ascribed to the system error with different approaches. Related discussion is included in the revised manuscript on Page 8 line 6-8.

Question 4: Will the thickness of the polymer thin film influence the device performance? How about the grafting amount of the Fc and TPA? Will they also influence the device performance? How about the solubility of PFTPA-Fc in other

organic solvents? Can some other environmental friendly solvents could be used to substitute toluene? These issues should be investigated carefully.

Answer: We thank the reviewer for his/her questions. Generally, the polymer film thickness will mainly influence the initial resistance of the polymer layer and the device ON/OFF ratio. Device with thicker polymer layer may show smaller HRS current and consequently higher ON/OFF ratio. Simultaneous control of the film thickness, compliance current and other factors may also change the switching behavior from bipolar and nonvolatile to volatile. This has been systematically investigated early by our and other groups (c.f., *Nanoscale* 2017-9-2449; *J. Mater. Chem. C* 2013-1-4858; *J. Phys. Chem. C* 2010-114-6117; *J. Phys. Chem. C* 2009-113-3855; *Nanotechnology* 2008-19-035203).

As replied in our previous revision, we used excess amount of ethynylferrocene during synthesis to react with the azide moieties of the PFTPA-N₃ precursors through “Click” chemistry. As confirmed by the NMR and FTIR data, a 100% “Click” reaction has occurred to achieve complete grafting of ferrocene moieties onto the pendant of PFTPA-Fc, giving each repeating unit of the PFTPA-Fc polymers containing two Fc unities and two TPA unities as shown in **Figure 1**. On the other hand, the quantity of grafted Fc moieties and the TPA groups will surely influence the electronic structure and electrical property of the polymer, because their electrochemical redox behavior are directly related to the chemical structure and composition of the material.

Since the primary aim of the present work is to conceptually demonstrate that polymer materials can be used to construct novel devices showing multistate memory and information processing capabilities, while full structure-property relationships between the polymer thin film thickness, the amount of Fc and TPA grafting and the device performance require extensive yet routine investigation, they are not included in the present work and will be explored in future study systematically. Nevertheless, related discussion is included in the Discussion section of the revised manuscript on page 17 line 21 to page 18 line 7.

For the solubility of PFTPA-Fc, beyond toluene, it can also be dissolved in THF and DMF to give a 10 mg/mL solution, as included in the previous revision Supplementary Figure 3. Chloroform and *N*-methyl pyrrolidone can also dissolve PFTPA-Fc.

On the other hand, other environmentally friendly solvents such as water and ethanol cannot dissolve PFTPA-Tc. As a matter of fact and stated in the Supplementary Information on page 17 line 5-7, the PFTPA-Fc was actually precipitated from the reaction mixture with alcohol and dialyzed with deionized water for purification. Related discussion on polymer solubility is included in the revised manuscript on page 6 line 13-15.

I certainly hope that this manuscript could be formally accepted for publication in *Nature Communications* this time. Thank you very much.

Best Wishes,

REVIEWERS' COMMENTS:

Reviewer #2 (Remarks to the Author):

The authors have nicely addressed most of my concerns and I am happy to recommend its publication in this form.